SOFTWARE

# Common data models to streamline metabolomics processing and annotation, and implementation in a Python pipeline

**Joshua M. Mitchell**[1], **Yuanye Chi**[1], **Maheshwor Thapa**[1], **Zhiqiang Pang**[2], **Jianguo Xia**[2], **Shuzhao Li**[1,3]*

1 The Jackson Laboratory for Genomic Medicine, Farmington, Connecticut, United States of America,
2 Institute of Parasitology, McGill University, Montreal, Quebec, Canada, 3 University of Connecticut School of Medicine, Farmington, Connecticut, United States of America

* shuzhao.li@jax.org

## Abstract

To standardize metabolomics data analysis and facilitate future computational developments, it is essential to have a set of well-defined templates for common data structures. Here we describe a collection of data structures involved in metabolomics data processing and illustrate how they are utilized in a full-featured Python-centric pipeline. We demonstrate the performance of the pipeline, and the details in annotation and quality control using large-scale LC-MS metabolomics and lipidomics data and LC-MS/MS data. Multiple previously published datasets are also reanalyzed to showcase its utility in biological data analysis. This pipeline allows users to streamline data processing, quality control, annotation, and standardization in an efficient and transparent manner. This work fills a major gap in the Python ecosystem for computational metabolomics.

**Data Availability Statement:** The version of the pcpfm, notebooks, and previously unreleased datasets used to generate the results presented in this manuscript are available at https://doi.org/10.

## Introduction

Metabolomics aims to comprehensively detect, identify, and quantify the diverse small molecules, i.e., metabolites, present in biological systems. This provides key information on biochemical phenotypes, often reflecting the function of genes and genomes. With the progress of technologies, metabolomics is becoming a regular component of many biomedical projects [1,2,3,4]. Thousands of metabolomics datasets are now available in major data repositories [5,6,7] and the annual citation of "metabolomics" in PubMed now exceeds ten thousand. Due to this increasing popularity, solutions for processing such data need to be better incorporated into the regular bioinformatics workflows [8,9,10]. This integration will require an ecosystem in both the R and Python programming languages, the two dominant languages for bioinformatics, each with unique strengths and a large user community.

The foundational tool of a software ecosystem in computational metabolomics is the preprocessing tool that, among other functions, converts raw data into feature tables representing signals of interest likely to represent metabolites. XCMS [11] has served this role for the R programming language, and various tools for further data processing, including annotation,

   

5281/zenodo.10629957. The HZV029 HILIC+, RP- are uploaded as study ID: ST003109, Two-Phase HILIC as study ID: ST003075 and HZV029 QC was already available as study ID: ST002233. The Checkmate dataset was retrieved from Metabolomics Workbench (https://www. metabolomicsworkbench.org) with study ID: ST00127. Bowen 2023 dataset was retrieved from Metabolights (https://www.ebi.ac.uk/metabolights/, accession code MTBLS2746). The Ansone 2021 dataset was retrieved from Metabolights (accession code MTBLS3852).

**Funding:** • This work was supported by the National Institutes of Health (U01CA235493, R01AI149746, R01AI149746S1, UM1HG012651 to SL). The funders had no role in study design, data collection and analysis, decision to publish, or preparation of the manuscript.

**Competing interests:** The authors have declared that no competing interests exist.

quality assurance and quality control (QA/QC), have been built utilizing its outputs [12,13,14,15]. Many optimization tools and pipelines have been built around XCMS [16,17,18,19,20]. Despite the popularity of Python in machine learning and bioinformatics in general, a robust ecosystem for metabolomics in Python remains lacking, primarily due to the lack of a preprocessing tool for metabolomics raw data. While a handful of Python tools have been developed over the past decade [21,22], they are either dated or not production-ready. With the recent release of Asari [23], a preprocessing tool implemented in Python, Python has become a viable option for data processing in computational metabolomics.

As computational metabolomics evolves, the community continues working to define operational terminology and best practices. These efforts have resulted in various workgroups and multiple publications [24,25,26,27]. Since metabolomics analysis is often part of larger biomedical projects, there is an urgent need to standardize terminologies that cover sample preparation, experimental protocols, steps of software processing and metadata. While Asari fills a key gap in the computational metabolomics ecosystem, the fundamental issue of interoperable data structures remains a challenge. To standardize the computational aspects of metabolomics analysis and empower future computational developments, a set of common, well-defined, and reusable data structures will be essential, regardless of the programming language. This paper, therefore, describes a collection of common data structures involved in metabolomics data processing and illustrates how they are utilized in a full-featured Python-centric pipeline.

## Design and implementation

Semi-automated data analysis pipelines are essential for the mainstream adoption of metabolomics and its continued growth in the biomedical sciences. With pipelines, researchers of diverse backgrounds can process their data quickly and meaningfully, allowing for higher throughput and more extensive experiments. Furthermore, pipelines allow researchers to define highly reproducible workflows that are repeatable and reproducible by others. Our pipeline, named the Python-centric pipeline for metabolomics (pcpfm), enables start-to-finish metabolomics data processing based on Asari. The pipeline ingests centroided mzML data or Thermo raw files and returns a human-readable set of tables summarizing the detected features and their annotations and sample metadata. Annotation is a major step after preprocessing, utilizing multiple sources, such as authentic compound libraries and tandem mass spectral libraries. Annotation levels in pcpfm are described in accordance with Schmanski 2014 [28]. Additionally, the pipeline performs various processing steps, including normalization, feature interpolation, removal of rare features, quality assurance, quality control evaluations, and generates PDF reports to summarize results.

We designed a set of core data models, which are described in the MetDataModel package and summarized in Table 1. The goal of MetDataModel is to encourage reuse and extension, therefore the data models are kept minimal. Developers are free to extend them to more detailed and specific models. Such extensions and applications are exemplified here in the pipeline package, pcpfm.

A mass spectrum typically consists of a list of m/z (mass to charge ratio) values and corresponding intensities. It can be from a full scan (MS$^1$) or tandem mass spectrometry (MS$^2$ and beyond). The mass spectrum can be in profile mode or centroid mode. In profile mode, the term "mass peak" is still used by some applications to refer to a group of m/z values that belong to the same ion species. Data in profile mode can be converted to centroid mode (mass peak picking) by software from the manufacturers or from scientific community, and usually done

**Table 1. Core concepts implemented in the MetDataModel package.**

| Name | Operational Definition |
|---|---|
| MS Spectrum | List of m/z values and associated intensity, typically from a scan on a mass spectrometer |
| Mass Track | An extracted ion chromatogram of consensus m/z, spanning the full retention time. |
| Elution Peak | Peak of intensity values along the axis of chromatography. |
| Feature | A set of peaks that are aligned across samples, specific to an experiment. |
| Empirical Compound | A group of associated features, typically isotopes and adducts, that belong to the same tentative compound and co-elute if there is chromatography. |
| Compound | A metabolite or a chemical of xenobiotic origin, including contaminants. |
| Reaction | Biochemical process that interconverts one or more compounds, often catalyzed by an enzyme. |
| Enzyme | A protein that catalyzes a biochemical reaction. |
| Gene | An inheritable sequence of nucleotides, some of which code for proteins. |
| Metabolic Pathway | A series of linked reactions that typically involve structurally related compounds, usually defined by human knowledge. |
| Metabolic Network | A set of reactions connected by shared compounds. Mathematically identical to pathway, but not limited by pathway definition. |
| Metabolic Model | A collection of metabolic reactions and their associated metabolites, enzymes, and genes. Additional parameters, e.g. reaction rates and flux rates, can be included. |
| Study | A collection of experiments on a set of related samples. |
| Experiment | A set of acquisitions collected on a set of samples using consistent methods. |
| Method | The approach and parameters used for data collection in an experiment, e.g., chromatography and ionization parameters. |
| Sample | A biological sample or a control sample that is analyzed in a study. A sample can be analyzed in multiple experiments, by a single or multiple methods. An instance of data file generated by analyzing a sample is referred to as an acquisition. Analytical replicates need to be modeled explicitly if used. |

by default in format conversion to the common mzML format. Centroided data is much reduced in size and there is little reason to use profile mode.

A mass spectrometer is often connected to chromatography (typically liquid phase or gas phase); therefore, such an experiment acquires many mass spectra at different chromatographic retention times. Thus, data processing requires the detection of signals across spectra, i.e., scans. Such signals are typically presented as an extracted ion chromatogram (EIC or XIC). In the Asari software, this concept of EIC is extended to a "mass track" [23], which is a vector of intensity values spanning the full scan range under one consensus m/z value. The use of mass tracks leads to new algorithms for alignment and feature detection [23]. Because "peak picking" or "peak detection" could refer to either mass peaks or elution peaks, we recommend the explicit term of elution peak detection. An elution peak is defined by ion intensity along the axis of retention time in the 2-dimensional representation. A mass peak is defined by ion intensity along the axis of m/z, usually in profile data. We define an elution peak at the level of a sample and as a feature at the level of an experiment (Table 1). The definition of "feature" here is consistent with its use in XCMS [11] and MZmine [29], but different from OpenMS [30]. OpenMS refers to a feature as a group of ions, likely due to its root in proteomics. The relationships between these concepts are illustrated in Fig 1A.

The relationship between metabolite, reaction, enzyme, gene, pathway, and network is described on right side of Fig 1A, which are collectively considered as a "metabolic model". Metabolic reactions are central to connect these entities, and the links to enzymes (proteins) and genes (measured in transcriptomics, genomics and epigenomics) are the most important basis for analyzing multi-omics data [31,32]. These concepts mirror the extensive development

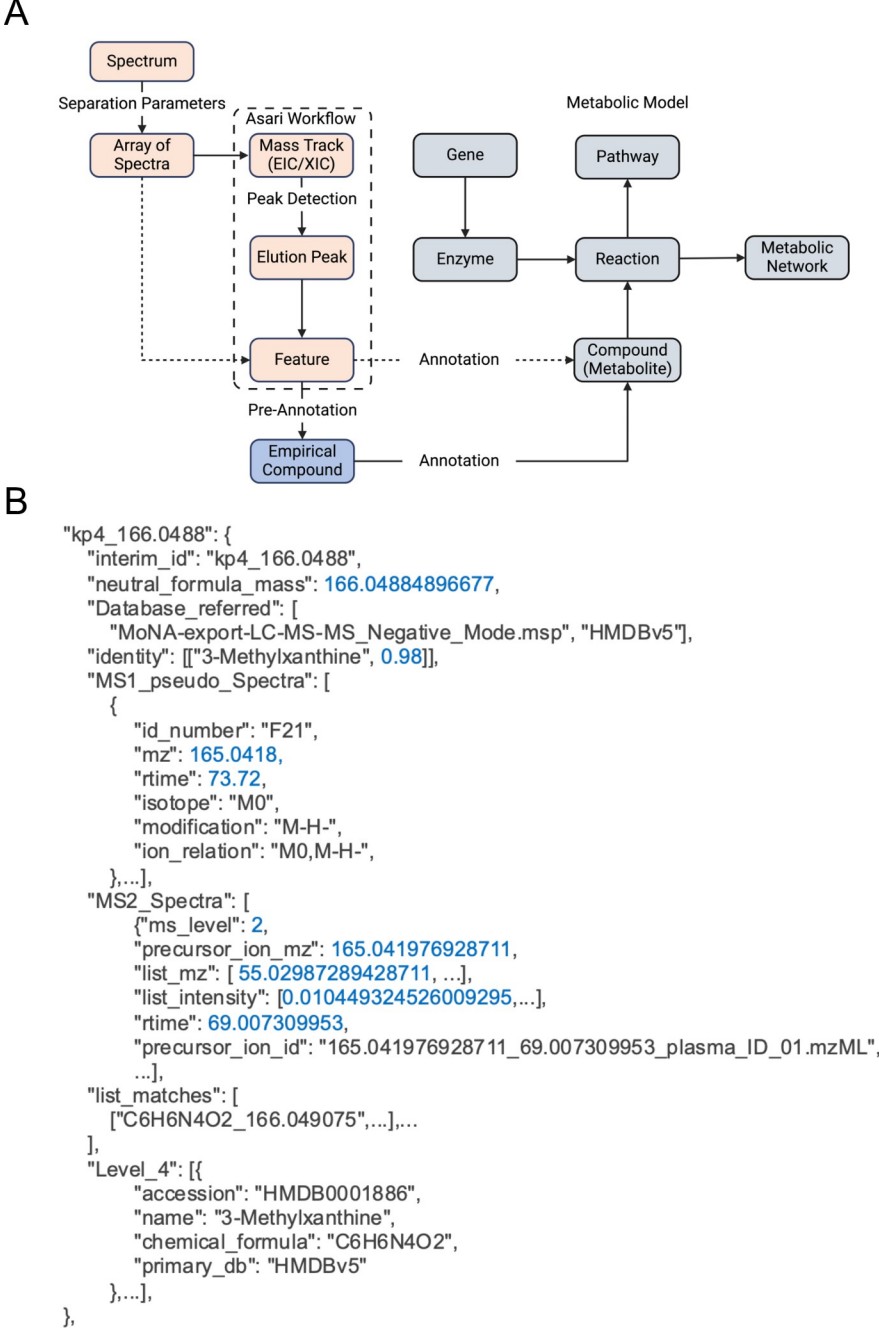

**Fig 1. Design of core concepts and data models in computational metabolomics. A)** The core concepts in MetDataModel with the metabolomics data processing in salmon and metabolic modeling in grey. We introduce "empirical compound" as a key bridge in between. The dashed lines indicate alternative workflows. Created with Biorender.com. **B)** Abridged empirical compound example including the listing of MS[1] features, annotation from MS[2] and other sources. This JSON format enables chaining of multiple annotation tools.

from the field of genome-scale metabolic models (GEMs) in the previous two decades. Connecting GEMs with the experimental measurement by mass spectrometry is not trivial, because a) the identifiers of metabolites need to be consistent; b) charge states of molecules and experimental measurements need to be consistent; c) a significant knowledge gap exists

between the GEMs and experimental metabolomics; and d) metabolite identification is limited in experimental metabolomics.

The reality of metabolomics is that many features are not definitively identified. We have introduced the concept of empirical compound to describe the measurement of a tentative metabolite (Fig 1B). For example, in LC-MS (liquid chromatography coupled mass spectrometry) metabolomics, some isomers (molecules of identical mass) may not be resolved, limiting the annotation level. That is, the isotopologues and adducts clearly belong to the group, but the group may be isomer A, isomer B, or a mixture of both. Empirical compounds model this property and serve as an operational unit to link computational steps. It has been part of the software implementation since version 2 of mummichog and version 4 of MetaboAnalyst [33]. This design enables an organized presentation of degenerate MS[1] features, and chaining annotation from MS[n] and multiple methods. The isotopes and adducts from pre-annotation are modeled as a grid structure, made computable by the khipu package [34], which is also incorporated in the pcpfm pipeline. Annotation remains the most critical step in the meaningful interpretation of metabolomics data and the field faces the challenge of handling annotation uncertainty and probability. Empirical compounds provide an operational data structure as a path forward.

The abstract concepts in Table 1 and Fig 1 are intrinsically agnostic to programming languages. We demonstrate their implementation in Python 3 and JSON in MetDataModel. The pcpfm package is written in Python 3 and JSON is used extensively for intermediary data. Many pipeline data structures inherit from, and expand upon, objects provided by the MetDataModel library. Specific extension of empirical compound is exemplified in Fig 1B.

The inputs to our pipeline minimally consist of .mzML or .raw files and a metadata CSV file, that minimally maps sample names to acquisition file paths. While Asari was initially developed for orbitrap data, pcpfm is expected to be compatible with the data from major manufacturers that can be converted into mzML format [35]. The final output consists of a feature table detailing the observed m/z and retention time values for observed features mapped to unique identifiers, an annotation table mapping these identifiers to annotations and metadata for those annotations, and a third table summarizing the acquisition and experiment-level metadata. This three-table format handles multiple annotations gracefully and will be supported in future versions of MetaboAnalyst and Mummichog for downstream analysis and interpretation.

Each step in an analysis corresponds to one command in the CLI and one function in the main pipeline process (S1 Table). In brief, every analysis starts with assembling an experiment object from the metadata and acquisition data. This experiment object records the location of intermediates on disk for reuse in later steps. Optionally, any.raw files are converted to centroided .mzML files using the ThermoRawFileParser [36] before preprocessing with Asari which yields a "preferred" and "full" feature table.

Quality control is necessary in every project but depends on the experimental design. Multiple QA/QC operations are available including PCA, t-SNE, correlation cluster maps, Z-scores that quantify the median pearson correlation to all other samples, the number of features or missing features per-sample are implemented as well as scatter plots that summarize median and mean feature intensities plus bar plots of TICs (total ion counts) at each step in the workflow are implemented. PCA and t-SNE are implemented as wrappers around scikit-learn functions [37], cluster maps use a combination of seaborn [38] for hierarchical clustering and either scipy [39] or numpy [40] for the calculation of the input correlation matrix. Z-scores and TICs are calculated using custom routines implemented using a mixture of pandas [41] and numpy. All plotting is based on matplotlib [42] except clustermaps which are using seaborn [38].

Operations to correct common data quality issues are provided including normalization, blank masking, batch correction, removal of uncommon features, and missing value imputation. Blank masking, missing value imputation, and feature removal is implemented using custom routines in Pandas and numpy, batch correction using a wrapper around pycombat [43], while normalization is implemented using numpy and can be performed in a one-pass or two-pass approach. In the two-pass approach, normalization is done within a batch of samples and then the batches are scaled to one another using their median TIC values based on the conserved features. Missing value imputation is implemented using a scalar multiple of the minimum observed intensity for that feature in the dataframe. Reasonable defaults for the user-provided parameters for these operations are implemented and described in S1 File while the ordering of steps is modifiable. For example, batch correction is sensitive to missing features and can be performed after removing frequently missing features or after missing value imputation; however the default workflow consisting of blank masking, outlier sample removal, normalization, outlier feature removal, imputation and filtering before annotation is recommended in that order to remove bias in each subsequent step. Outlier samples can be removed using any of the Z-scores or other 1-D metrics described above via a user-provided filter; however, by default, samples that have an absolute value for their number of features Z-score greater than 2.5 are dropped.

Empirical compounds are constructed from a feature table using Khipu [44] and most methods for empirical compounds concern annotation. Using MatchMS [45,46], $MS^2$ based annotations can be generated using data from DDA or deep scan workflows such as AcquireX [47,48] and $MS^2$ spectral databases such as MoNA [49] or authentic standards libraries. $MS^1$-based annotations are generated using our JSON metabolite services library and appropriately formatted inputs or m/z and retention time similarity to authentic standards. These annotations can be mapped back to any feature table to generate the previously mentioned tabular output. PDF reports can be created using the fpdf library [50]. The contents of the report can be defined by the end-user via a JSON template but by default include PCAs, log TICs, pearson correlation clustermaps, missing feature Z-score plots for each feature table, an accounting of all annotations and features explained for each set of empirical compounds created, a time-stamp for the report generation, and a timeline of all commands used in the analysis. Example reports are provided in S2 and S3 Files.

Most operations in the pipeline are chainable meaning they can be performed in a user-specified order with outputs from previous iterations being used as inputs. This flexibility allows users to build custom workflows; however, example workflows are provided as .sh and nextflow scripts [51]. Nearly all parameters are user-configurable, but reasonable defaults are provided and documented, allowing the pipeline to be as hands-off or hands-on as the end user desires.

## Results

The pcpfm is designed to prepare data for downstream data analysis, which can be performed by bioinformaticians or data scientists without a background in mass spectrometry. The major steps are shown in Fig 2A and a comparison of the provided functionality to other metabolomics data processing tools [19,20,52,53,54,55,56] is shown in S2 Table. Additionally, we demonstrate first the results on data processing, annotation, and quality control, then on biological applications. Seven metabolomics and one lipidomics datasets from four studies, three fully public [57,58,59] and one in-house, are used in these examples (details in S1 File).

A distinct advantage of pcpfm and Asari is the computational efficiency to process large datasets. The computational times are summarized on two high-resolution LC-MS datasets of

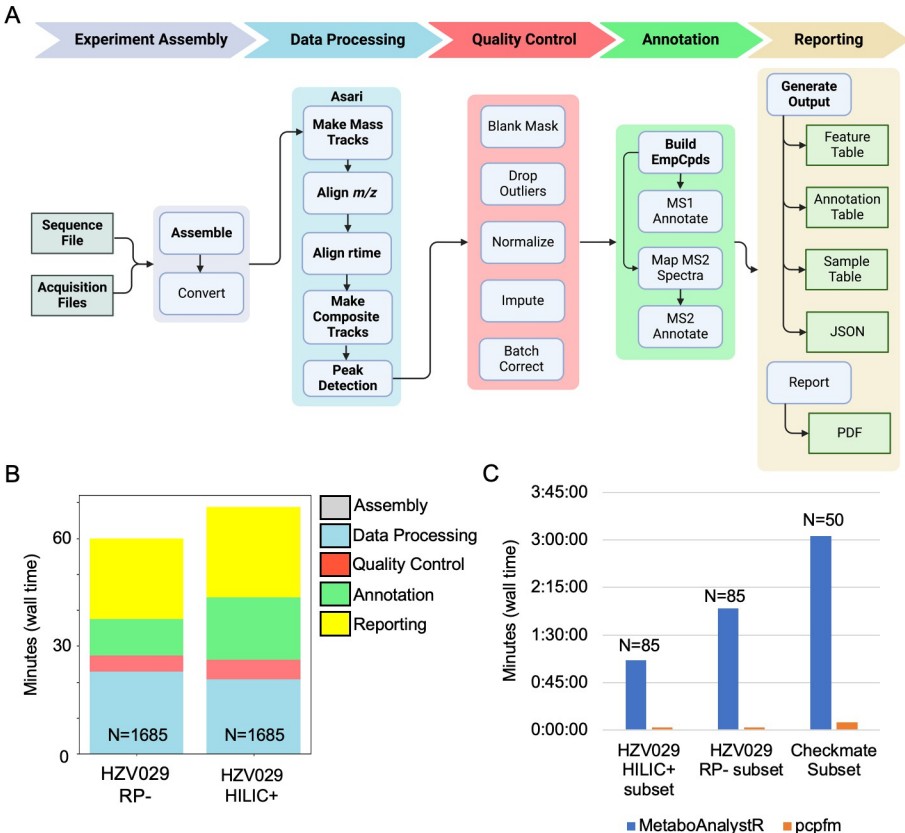

**Fig 2. Design and computational performance of the pcpfm pipeline. A)** The pipeline has five major sections: assembly, data processing, quality control, annotation and reporting. assembly creates the on-disk data structures needed for pcpfm analysis and optionally performs conversion to mzML. Data processing encapsulates everything from the start of a processing job to the creation of a feature table using Asari. Quality control consists of multiple chainable commands that allows for a raw feature table to be curated into a table suitable for downstream analysis. Annotation concerns the mapping of empirical compounds to metabolites using formula or MS$^2$ similarity to databases, m/z and retention time mapping to authentic standards and optionally, MS$^2$ similarity. Finally, reporting handles the creation of the three-table format for downstream analysis, PDF report generation, and JSON outputs for advanced users. Squares represent inputs and outputs, arrows represent dependencies between any steps, while bolded sections collectively represent a minimal workflow. Created with BioRender.com. **B)** Using the two largest datasets (N is the number of MS$^1$-only acquisitions), the high computational performance of our pipeline is demonstrated. Most of the wall time is spent during reporting. All steps are single threaded by default except Asari which uses 4 processes. In the HILIC+ and RP- datasets, 40008 and 32086 features are detected (full asari table including non-study samples) corresponding to 27851 and 23400 empirical compounds of which 16431 and 11962 received a level 4 annotation and 614 and 267 received a level 2 annotation. **C)** A comparison of the wall time required for a minimal pcpfm workflow (Asari+Khipu) compared to its MetaboAnalystR v4.0.0 equivalent on subsets of three studies where N is the number of MS$^1$-only acquisitions included in each subset. For the CheckMate subset, 3902 and 8907 features were detected by the MetaboAnalystR and PCPFM minimal workflows respectively while in HZV029 HILIC+ and RP- MetaboAnalystR workflow detects 2835 and 5966 features while the pcpfm workflow detects 12142 and 9939 respectively. All pcpfm counts are for the preferred feature table.

1685 samples. Processing and QC use less than half an hour on a laptop computer for each datasets, while the annotation step depends on the databases involved while report generation depends on the number of samples and intermediate tables selected for figure generation (Fig 2B). The computational performance of the pre-processing and pre-annotation was assessed by comparing a minimal pcpfm workflow consisting of pre-processing and pre-annotation only using asari and khipu respectively versus a MetaboAnalystR v4.0.0-based pipeline (using OptiLCMS v1.1.0's implementation of XCMS, and CAMERA [13], details in S1 File) on

a subset of three datasets (Fig 2C), showing a clear improvement in the performance of our pipeline.

The metabolomics community have a consensus that metabolite annotation should be reported according to its confidence level. We have incorporated empirical compounds into both $MS^1$ and $MS^2$ annotations. By building empirical compounds first, i.e. pre-annotation via the khipu package, $MS^1$ annotation is improved because the search of databases does not query many degenerate features (Fig 3A). The $MS^2$ annotation utilizes MatchMS but with an optimization using an interval tree algorithm [60]. Because there are many implementations of $MS^2$ annotation under similar principles, it is important to be explicit on the algorithm in pcpfm (Fig 3B). The $MS^2$ annotation in pcpfm is efficient enough to run large experiments on consumer-grade hardware, as shown in Fig 2B. When authentic compounds are used to annotate metabolites, it is straight forward to match their m/z and retention time to biological samples (Fig 3C). Multiple annotations of different sources are chained in the empirical compound data structure (Fig 1B), which is amendable to future enhancements, e.g., context specific databases.

We compared the $MS^2$ annotations generated by the pcpfm to those from vendor's software, Compound Discoverer (CD) [48]. Full details for the annotation procedures in both softwares are provided in S1 File; however, CD annotations did require an additional step to map the generated annotations to the Asari feature tables which used an m/z tolerance of 10 ppm and a retention time tolerance of 30 seconds. For both pcpfm and CD annotations sets of annotated features were constructed by concatenating the annotated compound name with the asari feature (e.g., Caffeine_F1345) and these annotation sets then compared using set operations in Python. Considerable overlap is seen between CD and pcpfm annotations (Fig 3D). Because the algorithm in CD is closed source, it is not feasible to trace the differences between the tools, which highlights the importance of open-source tools for continued improvement.

The applications of pcpfm to quality control are demonstrated on a dataset consisting of 17 batches and 1685 samples (Fig 4). This analysis was performed using a batch-correction variation of our default workflow (S1 File). First, the QC metrics generated by Asari are summarized using kernel density plots to illustrate the high quality of features yielded by asari as evidenced by their high cSelectivity, peak shape (i.e., goodness of fit to a gaussian), their peak areas and high signal-to-noise ratio (Fig 4A). Next, hierarchical clustering of the inter-sample pearson correlation across all features was performed revealing two clusters of samples, representing a clear batch effect that was traced back to a recalibration of the instrument after batch 8 (Fig 4B). The log TICs of a random subset of samples and PCA plots were generated to further investigate these batches and identify abnormal samples (Fig 4C and 4D). The standard two-pass normalization does not adaquetly correct the batch effect; however, after normalization and batch correction via pycombat, both the log TICs and PCA show more consistency and no sub-clustering, suggesting the batch effect was largely mitigated.

Another common data quality issue addressed by the pcpfm are failed injections. Using the per-sample number of features Z-score, we demonstrate the ability to detect failed injections automatically in two datasets (Fig 4E). Failed injections are readily identified by their anomalously low Z-score (red) compared to successful injections (black). When the failed injections are compared to the preceeding successful injection, the absence of clear signal is appreciated in their TICs, confirming they were failed injections (Fig 4E). These results motivated the inclusion of this metric and a default cutoff of $|Z| > 2.5$ for the removal of outliers by default in the pcpfm.

Multiple previously published datasets were reanalyzed using pcpfm to evaluate the pipeline's general suitability. These analyses were performed using either a modified default

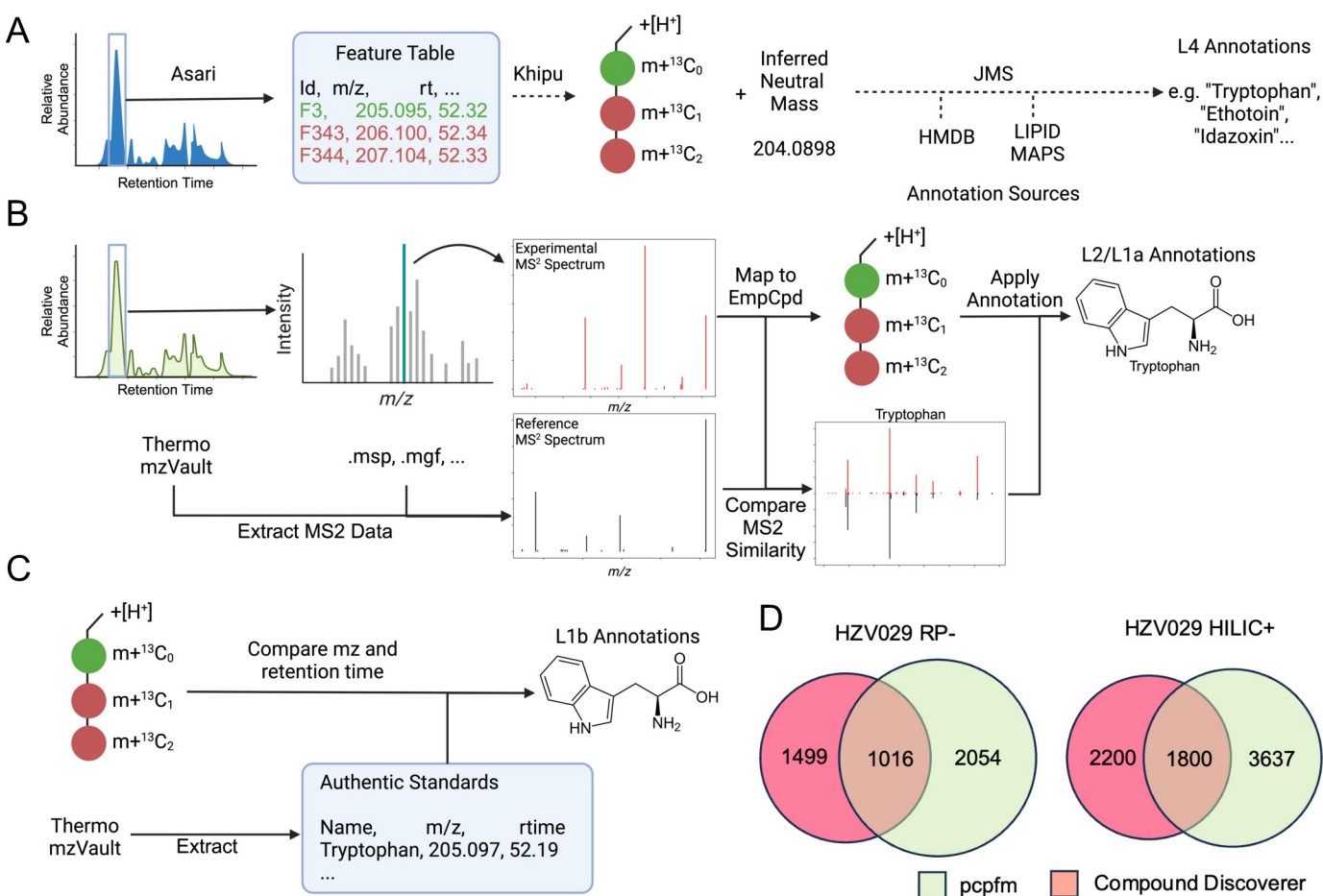

**Fig 3. Annotation methods in pcpfm. A)** Empirical compounds are constructed from Asari feature tables using khipu, which groups degenerate features such as isotopologues and adducts. The inferred neutral mass of an empirical compound is compared to known metabolites to generate level 4 annotations (via JMS, https://github.com/shuzhao-li-lab/JMS). Panels A, B, and C created with BioRender.com. **B)** Level 2 and 1a annotations are generated using MS$^2$ similarity. Experimental MS$^2$ spectra are mapped to empirical compounds and then compared to reference spectra, to annotate metabolite structures. **C)** Level 1b annotations are generated based on m/z and retention time match to authentic chemical standards. The use of empirical compound improves search efficiency and reduces false positives, while annotations at all levels can also be mapped to the feature level. **D)** Overlap of MS$^2$ annotations by pcpfm and CD in the two HZV029 plasma datasets. Detailed dissection of the differences is difficult since CD is closed-source.

workflow or, in the case of the CheckMate analysis, a minimal workflow as described in S1 File. Bowen et al (2023 [57]) designed a specialized xenobiotic-focused workflows to detect metabolites of the drug sunitinib. Our pipeline with default parameters detects all but one of the previously reported sunitinib-related metabolites in cardiomyocyte cell pellets and all features in culture media (Fig 5A) based on a 5 ppm m/z tolerance and 10 second retention time tolerance to the features reported in Bowen et al 2023. The sole missing feature is due to low signal-to-noise ratio, not passing Asari quality threshold (Fig 5B). Using ANOVA followed by Benjamini-Hochberg correction [61], sunitinib-treated and control cell pellets were compared and hierarchical clustering performed using the significant features (Fig 5C). This yields two distinct clusters corresponding to the treated and control samples consistent with an induced metabolic response resulting from sunitnib exposure as reported in the original analysis [57]. These results indicate the potential of pcpfm as a simplified yet broadly applicable workflow.

To compare pcpfm feature detection against a state-of-art R-based pipeline (MetaboAnalystR v4.0.0), we reprocessed a subset of published metabolomics data on the CheckMate

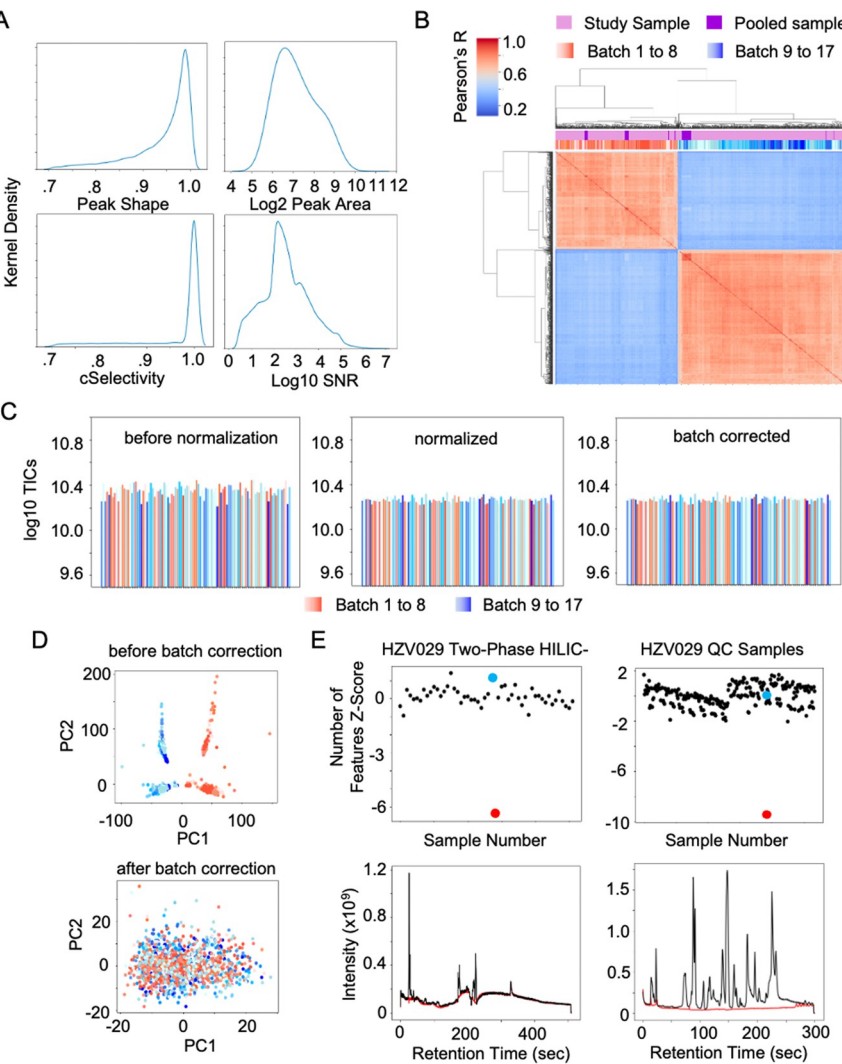

**Fig 4. Examples of quality control in the pcpfm pipeline. A)** A collection of QA/QC metrics generated by Asari on an example dataset ("HZV029 Plasma RP-"). **B)** The correlation clustermap of all study samples and pooled samples from the HZV029 Plasma RP- dataset (preferred feature table) illustrating the batch effect induced by instrument calibration. **C)** Log10 TICs of a random subset of samples before normalization, after normalization, and after batch correction. **D)** PCA demonstrating the presence of a batch effect (top) and its removal (bottom). **E)** Detection of failed acquisition by the number of feature Z-scores. The failed injection is highlighted in red and a representative "good" injection in blue for both the plasma HZV029 Two-Phase HILIC- and HZV029 QC dataset (left and right, top). The two-phase failed injection is simulated by replacing a missing sample with an empty vial while the other was identified post-hoc. The TICs of the failed and good injections are shown in red and black respectively (bottom).

immunotherapy cohort [58] using comparable minimal workflows as previously described (additional details in S1 File). The authors' in-house metabolite library serves as a proxy of ground truth here. An m/z tolerance of 5 ppm and an RT tolerance of 5 seconds was used to identify library features in the MetaboanalystR and pcpfm results.The sets of identified features were then compared to ground truth using the set logic operations in Python. The pcpfm consistently detects more features representing more true metabolites than the MetaboanalystR workflow, however, considerable overlap is observed (Fig 5D).

Lastly, as an example for generating biologically meaningful results, we reanalyzed the metabolomics data from a COVID-19 exposure and recovery cohort (Ansone 2021, [59]).

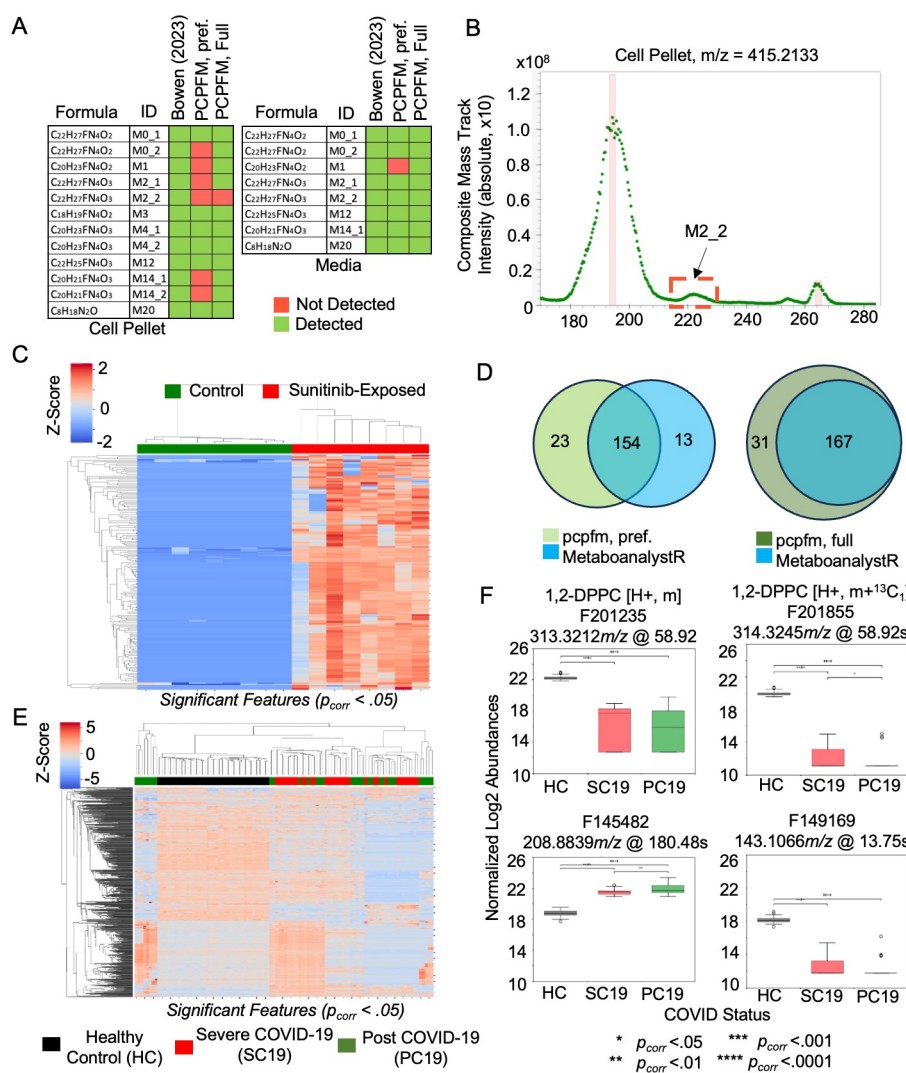

**Fig 5. Applications of pcpfm to analyzing biological datasets. A)** In the Bowen 2023 cardiomyocyte dataset, the pcpfm identifies most of the reported sunitinib-related features in both cell pellets and media using a standard workflow. Asari and pcpfm output both a preferred feature table and a full feature table, the former of higher feature quality and the latter more inclusive. **B)** The mass track for the sole feature undetected in the Bowen 2023 cell dataset is shown and the suspected undetected peak is in red box (M2_2), which fails to pass Asari's quality requirement. **C)** Significant differential metabolite features between sunitinib exposure groups in cell pellets. ANOVA p-values are corrected for multiple testing by Benjamini-Hochberg method. **D)** Both the pcpfm and MetaboAnalystR were used to extract features from a subset of the CheckMate study. Of 202 compounds in their authentic standard library, MetaboAnalystR identified 167, while the full table from the pcpfm identified 198 of the confirmed features. **E)** Clustering pattern of the Ansone 2021 cohort using features differentially abundant between treatment groups. **F)** Example boxplots of differentially abundant features in the Ansone 2021 cohort. F201235 and F201855 (top) were mapped to the same empirical compound that was tentatively annotated as 1,2-DPPC, a pulmonary surfactant by its sole level 4 annotation. Significance was evaluated using ANOVA and post-hoc Tukey's HSD test in E and F.

Following pcpfm, the significant features tested by ANOVA followed by a Tukey's HSD [62] were subjected to hierarchical clustering (Fig 5E), which recapitulated the original observation that metabolic profiles cluster by COVID infection and recovery vs. control in the Anosne 2021 paper. The box plots of selected features confirm the patterns of abundance changes in participant groups (Fig 5F). Interestingly, two features (Fig 5F, top) are found to belong to an empirical compound with a single level 4 annotation to 1,2-dipalmitoylphosphatidylcholine

(1,2-DPPT), a pulmonary surfactant known to be less abundant in COVID patients than healthy controls [63]. This result demonstrates that novel biology can be gained with the pcpfm. The Jupyter notebooks and workflows underlying these examples are included in the pcpfm code repository, so that users can easily perform their own data analysis based on the templates.

## Availability and Future Directions

The MetDataModel and pcpfm are available through GitHub (https://github.com/shuzhao-li-lab/metDataModel and https://github.com/shuzhao-li-lab/PythonCentricPipelineForMetabolomics), and both are installable by pip via PyPi or from source. All dependencies are open source and downloadable via pip, except for the Thermo-RawFileConverter and mono framework, both of which are optional. Example workflows are provided in bash and as nextflow; however, users can implement their own using the CLI or the pipeline internals available using standard Python conventions for APIs. API usage will be officially supported in an upcoming release.

Future development of pcpfm will implement additional options and methods for data processing, including normalization, interpolation, and batch correction. Improving support for non-orbitrap instruments is another priority for the pipeline and the underlying Asari algorithm. A cloud-based application is planned to allow users to process data in a friendly web interface.

## Supporting information

**S1 Table. List of commands in the pcpfm pipeline, their inputs and outputs, and if they are chainable.**
(XLSX)

**S2 Table. Comparison of pcpfm features to other metabolomics data processing tools.**
(XLSX)

**S1 File. Description of datasets, methods for generating previously unpublished datasets, and compound discoverer annotation workflow.**
(PDF)

**S2 File. HZV029 Plasma HILIC+ example PDF report.**
(PDF)

**S3 File. HZV029 Plasma RP- example PDF report.**
(PDF)

**S4 File. zip of pcpfm v1.0.13.**
(ZIP)

**S5 File. zip of MetDataModel v0.6.1.**
(ZIP)

## Acknowledgments

We would like to thank Paul Robson, Arti Taggar, Julianna Alcoforado Diniz, Zukai Liu, and Lucas Chang who graciously provided data for the initial development and testing of the pipeline.

## Author Contributions

**Conceptualization:** Shuzhao Li.

**Data curation:** Maheshwor Thapa.

**Formal analysis:** Joshua M. Mitchell.

**Funding acquisition:** Shuzhao Li.

**Investigation:** Joshua M. Mitchell, Yuanye Chi, Shuzhao Li.

**Methodology:** Joshua M. Mitchell, Maheshwor Thapa, Zhiqiang Pang, Shuzhao Li.

**Software:** Joshua M. Mitchell, Yuanye Chi, Shuzhao Li.

**Supervision:** Jianguo Xia, Shuzhao Li.

**Validation:** Yuanye Chi, Zhiqiang Pang.

**Visualization:** Joshua M. Mitchell.

**Writing – original draft:** Joshua M. Mitchell.

**Writing – review & editing:** Joshua M. Mitchell, Zhiqiang Pang, Jianguo Xia, Shuzhao Li.

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
