## [Decision Letter · Decision Letter 0]

8 Mar 2024

Dear Dr. Li,

Thank you very much for submitting your manuscript "Common data models to streamline metabolomics processing and annotation, and implementation in a Python pipeline" for consideration at PLOS Computational Biology.

As with all papers reviewed by the journal, your manuscript was reviewed by members of the editorial board and by several independent reviewers. In light of the reviews (below this email), we would like to invite the resubmission of a significantly-revised version that takes into account the reviewers' comments.

As is clear from all reviewer comments, the reviewers appreciated the Python-centric focus with this method. Current metabolomics analysis/interpretation workflows lean towards R, but a very large user base now works with Python based code. Overall, the current manuscript is also clear in its goals as well as outcomes. However, there are short comings in how the comparisons/benchmarking has been done (compared to existing metabolomics processing/annotating pipelines). There are also several sections in the manuscript which are unclear (in terms of what exactly was done, or QA/QC benchmarks), or are missing important details. This requires careful, informative revision.

We cannot make any decision about publication until we have seen the revised manuscript and your response to the reviewers' comments. Your revised manuscript is also likely to be sent to reviewers for further evaluation.

Sincerely,

Sunil Laxman, PhD

Academic Editor

PLOS Computational Biology

Mark Alber

Section Editor

PLOS Computational Biology

As is clear from all reviewer comments, the reviewers appreciated the Python-centric focus with this method. Current metabolomics analysis/interpretation workflows lean towards R, but a very large user base now works with Python based code. Overall, the current manuscript is also clear in its goals as well as outcomes. However, there are short comings in how the comparisons/benchmarking has been done (compared to existing metabolomics processing/annotating pipelines). There are also several sections in the manuscript which are unclear (in terms of what exactly was done, or QA/QC benchmarks), or are missing important details. This requires careful, informative revision.

Reviewer's Responses to Questions

**Comments to the Authors:**

Reviewer #1: In this paper, the authors propose a Python-centric pipeline for (tandem) LC-MS data processing and annotation. The pipeline consists of the pcpfm and metDataModel packages, providing an interface for subsequent Python analysis. This paper is well written and organised. The software is useful in metabolomics data analysis.

1. I would suggest the authors provide more detailed descriptions of each step in their pipeline. For example, how does the pipeline detect and remove outliers? What QA/QC measures are implemented? What missing value imputation approaches are used? In addition, the authors should clarify their contributions by specifying whether they implemented the algorithms, e.g., t-SNE themselves or used a function from (which?) existing Python packages but wrapped for LC-MS data.

2. The authors should provide a clear justification for their workflow. For instance, explaining why batch correction is done after missing value imputation. What "various statistical tests" are used, and how do they "quantify sample properties"?

3. What does the author mean by saying "Some operations can be made batch aware ..." and "These operations are implemented using a mixture of sklearn (please refer to the package as scikit-learn) ..."?

4. The author should clearly explain the outputs of the pipeline. For example, what are the contents of the "PDF reports"? What results from each step in Fig 2 will be included?

5. The url of the MetDataModel GitHub repo is correct but the link is incorrect.

Reviewer #2: The authors present a Python-based pipeline for end-to-end analysis of LC-MS/MS metabolomics data. The pipeline brings together multiple existing tools that perform tasks including feature extraction, quality control/alignment, and annotation. The authors additionally propose a set of data structures with the aim of promoting consistency and interoperability between different metabolomics data processing software.

The primary strength of this work lies in the integration of existing tools that handle the different elements of typical metabolomics data processing, making complex data analysis tasks more tractable . The Python-centric implementation also adds to the appeal of this work, as both Python and R have significant investment from the scientific community but existing metabolomics data processing software leans toward R.

The primary weakness of this work comes from a lack of appropriate discussion of the methodology and results. The authors present results from analyzing multiple metabolomics/lipidomics datasets, which include some critical comparisons to other existing software tools, but description of methodological details and results are lacking. It is my opinion that appropriate rigor in the presentation of methodology and results is requisite for the impact of this work to be convincingly conveyed. See major points below for more specific examples.

Major Points:

- (line 193) There is no reference information for the four studies mentioned. This is necessary for the reader to be able to compare previous results with those from the presented software.

- (line 199) There is no methodological description of how the test was conducted. How is the reader meant to believe that the performance comparison to XCMS is fair/valid/informative?

- (line 200) It is not adequately explained what Figure 2C is depicting (e.g. what is N?).

- (lines 195-200) The discussion of results is lacking important details, such as how many features are detected from either software, how many are annotated, what level of confidence are the features annotated with?

- (lines 214-220) There is no appreciable description or discussion of the methodology or results presented.

- (lines 221-231) There is no appreciable description or discussion of the methodology or results presented. This section seems to highlight the conclusions without describing the results themselves or how those results support the derived conclusions.

- My feedback for the results sections referring to Figure 5 are likewise to the comments above

Minor Points:

- (line 15) Typo in abstract: “… it is essential is have a set …”

- In my view, the implementation of the MetDataModel classes would benefit from two changes: 1) use of dataclasses and 2) use of type annotations. Dataclasses (see PEP 557) are a better construct for the intended use case as I understand it. Semantically they convey the intent of the defined objects as means of storing state with minimal behavior. Syntactically they greatly simplify the class definitions since only the desired attributes (with default values where applicable) and methods (seems like serialize is the main one) must be explicitly defined while boilerplate methods like __init__, __str__, __repr__, etc. are defined for you. Dynamic typing is a convenient feature of Python, but the reality is that there are also many other languages in common use which conform to more strong typing conventions. With the stated goal of defining these data structures in a way that is language agnostic and generalizable, I feel that the inclusion of type annotations for the attributes of the MetDataModels classes is a must in order to avoid ambiguity for those who wish to adapt the proposed data model into applications written in strongly typed languages.

Reviewer #3: The presented article aims to fill gaps in metabolomics processing in Python environment. It provides several vital data processing steps from raw data to data table with identification and annotation.

Although I agree that MetaboAnalyst is the most popular and convenient software tool in metabolomics, you need to provide a comparison of functionalities with some other software based on R and with or without GUI.

First of all there are patRoon (doi: 10.1186/s13321-020-00477-w ; https://github.com/rickhelmus/patRoon) and xcmsrocker/rmwf (doi: 10.1186/s13321-022-00586-8 ; https://github.com/yufree/xcmsrocker ; https://github.com/yufree/rmwf). They mainly focus on comprehensive raw data processing, peak detection and annotation, including MS2.

Also OUKS provides an extensive variety of options for peak table processing such as: correction, normalization, imputation, statistical analysis and QA/QC (doi: 10.1021/acs.jproteome.1c00392; https://github.com/plyush1993/OUKS).

Add comparison with GUI tools such as Web-version of MetaboAnalyst, Workflow4Metabolomics (doi: 10.1016/j.biocel.2017.07.002 ; https://workflow4metabolomics.org/) and IP4M (doi: 10.1186/s12859-020-03786-x).

**Have the authors made all data and (if applicable) computational code underlying the findings in their manuscript fully available?**

Reviewer #1: Yes

Reviewer #2: Yes

Reviewer #3: Yes

PLOS authors have the option to publish the peer review history of their article (what does this mean?). If published, this will include your full peer review and any attached files.

Reviewer #1: No

Reviewer #2: No

Reviewer #3: No

Figure Files:

Data Requirements:

Please note that, as a condition of publication, PLOS' data policy requires that you make available all data used to draw the conclusions outlined in your manuscript. Data must be deposited in an appropriate repository, included within the body of the manuscript, or uploaded as supporting information. This includes all numerical values that were used to generate graphs, histograms etc.. For an example in PLOS Biology see here: http://www.plosbiology.org/article/info:doi%2F10.1371%2Fjournal.pbio.1001908#s5.
---

## [Decision Letter · Decision Letter 1]

20 May 2024

Dear Dr. Li,

We are pleased to inform you that your manuscript 'Common data models to streamline metabolomics processing and annotation, and implementation in a Python pipeline' has been provisionally accepted for publication in PLOS Computational Biology.

Best regards,

Sunil Laxman, PhD

Academic Editor

PLOS Computational Biology

Alison Marsden

Section Editor

PLOS Computational Biology

Reviewer's Responses to Questions

**Comments to the Authors:**

Reviewer #1: I appreciate the authors' efforts to improve the quality of the paper. They have responded properly to the previous comments. My only suggestion is to improve the design for Figure 2 A.

Reviewer #2: I am satisfied with the revisions.

Reviewer #3: -

**Have the authors made all data and (if applicable) computational code underlying the findings in their manuscript fully available?**

Reviewer #1: Yes

Reviewer #2: None

Reviewer #3: Yes

PLOS authors have the option to publish the peer review history of their article (what does this mean?). If published, this will include your full peer review and any attached files.

Reviewer #1: No

Reviewer #2: No

Reviewer #3: No

---

## [Editor Report · Acceptance letter]

31 May 2024

PCOMPBIOL-D-24-00256R1 

Common data models to streamline metabolomics processing and annotation, and implementation in a Python pipeline

Dear Dr Li,

I am pleased to inform you that your manuscript has been formally accepted for publication in PLOS Computational Biology. Your manuscript is now with our production department and you will be notified of the publication date in due course.

With kind regards,

Anita Estes
